# The Role of Genetic Testing in Children Requiring Surgery for Ectopia Lentis

**DOI:** 10.3390/genes14040791

**Published:** 2023-03-25

**Authors:** Mohammud Musleh, Adam Bull, Emma Linton, Jingshu Liu, Sarah Waller, Claire Hardcastle, Jill Clayton-Smith, Vinod Sharma, Graeme C. Black, Susmito Biswas, Jane L. Ashworth, Panagiotis I. Sergouniotis

**Affiliations:** 1Eye Department, St. James’s University Hospital, Leeds LS9 7TF, UK; 2Manchester Royal Eye Hospital, Manchester M13 9WL, UK; 3Division of Evolution, Infection and Genomics, School of Biological Sciences, University of Manchester, Manchester M13 9PL, UK; 4Manchester Centre for Genomic Medicine, St. Mary’s Hospital, Manchester M13 9WL, UK; 5Institute of Biochemistry and Molecular Genetics, Faculty of Medicine, University of Ljubljana, LJ1000 Ljubljana, Slovenia

**Keywords:** ectopia lentis, genetic testing, Marfan syndrome

## Abstract

Non-traumatic ectopia lentis can be isolated or herald an underlying multisystemic disorder. Technological advances have revolutionized genetic testing for many ophthalmic disorders, and this study aims to provide insights into the clinical utility of genetic analysis in paediatric ectopia lentis. Children that underwent lens extraction for ectopia lentis between 2013 and 2017 were identified, and gene panel testing findings and surgical outcomes were collected. Overall, 10/11 cases received a probable molecular diagnosis. Genetic variants were identified in four genes: *FBN1* (associated with Marfan syndrome and cardiovascular complications; *n* = 6), *ADAMTSL4* (associated with non-syndromic ectopia lentis; *n* = 2), *LTBP2* (*n* = 1) and *ASPH* (*n* = 1). Parents appeared unaffected in 6/11 cases; the initial presentation of all six of these children was to an ophthalmologist, and only 2/6 had *FBN1* variants. Notably, 4/11 cases required surgery before the age of 4 years, and only one of these children carried an *FBN1* variant. In summary, in this retrospective cohort study, panel-based genetic testing pointed to a molecular diagnosis in >90% of paediatric ectopia lentis cases requiring surgery. In a subset of study participants, genetic analysis revealed changes in genes that have not been linked to extraocular manifestations and highlighted that extensive systemic investigations were not required in these individuals. We propose the introduction of genetic testing early in the diagnostic pathway in children with ectopia lentis.

## 1. Introduction

Ectopia lentis is the dislocation of the crystalline lens from its normal position in the anterior segment of the eye. Dislocation can be partial or complete and may be primary or secondary to trauma or other factors [1]. The complications of ectopia lentis depend on the degree and the direction of the displacement. Sequalae include glaucoma, retinal detachment and, in young children, amblyopia. Notably, early identification and management with optical and/or surgical correction results in better visual outcomes [2].

Ectopia lentis in the paediatric population is often sporadic, although Mendelian inheritance patterns are observed in a proportion of affected families [1,3]. It can present as an isolated finding, or it can be associated with other ocular developmental abnormalities such as aniridia [1,3]. Alternatively, it can be a manifestation of a multisystemic condition such as Marfan syndrome or, less commonly, homocystinuria, Weill-Marchesani syndrome or sulphite oxidase deficiency [3]. To date, at least 14 genes have been implicated in the development of ectopia lentis and its associated systemic conditions (including *AASS*, *ADAMTS10, ADAMTS17, ADAMTSL4, ASPH, BCOR, CBS*, *COL18A1*, *FBN1*, *LTBP2*, *P3H2, PORCN, SUOX* and *VSX2*).

Given the heterogenous aetiology and the frequent systemic implications, several investigative techniques are routinely used in the assessment of children with ectopia lentis. Amongst them, genetic testing has been shown to have a high diagnostic yield in affected individuals [4]. Despite this, the variation in the current provision of genetic testing remains significant. 

Here, we report genetic findings and surgical outcomes in a cohort of children that underwent surgery for non-traumatic ectopia lentis. The clinical utility of genetic testing is highlighted, and insights into the aetiology of lens dislocation in this age group are provided. 

## 2. Materials and Methods

Study participants were retrospectively recruited through the paediatric surgical database at Manchester Royal Eye Hospital, UK. The criteria for inclusion in the study were: (i) age of 16 years or less; (ii) a diagnosis of unilateral or bilateral non-traumatic ectopia lentis; (iii) surgical lensectomy performed between 1 November 2014 and 1 November 2017. Cases managed conservatively were excluded. The genetic and, to an extent, the clinical findings from 7 affected probands have been previously reported [5,6].

The case notes and electronic healthcare record entries of all study participants were inspected, and relevant clinical data were systemically collected. In brief, documented encounters with clinical genetics, ophthalmology and other hospital services were reviewed, and the relevant data were captured in a spreadsheet application. Ocular and extraocular findings and information on visual function, surgical outcomes and relevant family history were recorded. 

All study participants underwent clinical genetic testing at the North West Genomic Laboratory Hub, Manchester, UK. Proband DNA was analyzed to look for alterations in up to 114 genes previously associated with lens abnormalities. Details on the panel-based tests that were used for genomic analysis can be found in [5]. One proband (case 7) underwent whole genome sequencing and virtual gene panel analysis as described in [6].

## 3. Results

Eleven paediatric patients met the inclusion criteria for this study, four male and seven female. The median age at presentation was 3 years (range 0–5), and the median age when the first lens procedure was undertaken was 5 years (range 1–8). Study participants were not knowingly related, and all had bilateral ectopia lentis. 

A history of parental consanguinity was noted in two probands. The majority of patients had European ancestries (6/11 cases, 55%). Family history of Marfan syndrome or ectopia lentis was recorded in 5/11 cases (46%). Most study participants (8/11 cases, 73%) initially presented to ophthalmic services prior to any other relevant medical specialty. The remaining three children (27%)—who had input from a different clinical service in a hospital setting before having an ophthalmology review—were all found to carry genetic variants in the *FBN1* gene. Presenting features included iris “flickering” (3/11), reduced vision (3/11), high myopia (1/11) and a high-arched palate at a postnatal screening assessment (1/11); 2/11 probands were referred via community screening due to a family history of Marfan syndrome (Table 1). 

Bilateral sequential or consecutive surgical lensectomy was performed in all patients within a median of 2 years of presentation to ophthalmology services. It was noted that 5/11 cases (45%) required surgery before the age of 4 years. Lensectomy was performed through an anterior approach in 9/11 cases. In the remaining 2 cases, a posterior (pars plana) technique was used. The surgeon was at consultant (9/11) or clinical fellow (2/11) grade. All patients were left aphakic postoperatively and received correction with glasses (10/11) or contact lenses (1/11). 

The pre-operative median visual acuity was 0.54 LogMAR. There was significant improvement post lensectomy surgery, and the median best corrected vision at the most recent assessment was 0.22 LogMAR. It was noted that the visual acuity decreased in both eyes in one patient (case 5). These findings are shown in Figure 1 and Table 2.

Additional ocular features were detected in 8/11 participants (73%); these included megalocornea, ectopia pupillae, spherophakia, elevated intraocular pressure and, most frequently, high myopia with astigmatism (Table 3). Although most probands were myopic, two children where hypermetropic. One affected child (case 11) had significant anisometropia (32.25 spherical equivalent). The median refractive error for the cohort pre-operatively was −4.25 (range −20.75 to +12.00 spherical equivalent); post-operatively, this was +12.825 (range +4.875 to +21.75 spherical equivalent). There was a significant reduction in the degree of astigmatism post-operatively.

All study participants had documented assessment by a consultant clinical geneticist. Extraocular abnormalities were recorded in 6/11 cases (55%); these included skeletal malformations (such as flat feet, arachnodactyly, pectus carinatum), high arched palate and cardiovascular abnormalities (patent ductus arteriosus and fenestrated atrial septal defect). Cardiovascular abnormalities were noted in two participants, and both of these cases had relevant family history.

A molecular diagnosis was made in 10/11 cases (diagnostic yield of 91%). Heterozygous variants in the *FBN1* gene were detected most frequently (*n* = 5), followed by biallelic variants in *ADAMTSL4* (*n* = 2), *LTBP2* (*n* = 2) and *ASPH* (*n* = 1). The detected genetic changes are shown in Table 3. Findings of interest included a heterozygous *de novo* variant in *FBN1*, c.356G > A (p.Cys119Tyr), that was detected in a 3-year-old patient (case 6). This led to the initiation of formal cardiology screening in this proband. In two cases (case 1 and 3), biallelic *ADAMTSL4* changes were detected, resulting in safe discharge from cardiology services. The *ADAMTSL4* gene is not associated with extraocular features, but both of these probands had a relatively severe ocular phenotype and required surgical intervention before the age of 3 years. Two patients were homozygous for *LTBP2* variants: case 4, who carried the c. 3427delC (p.Gln1143ArgfsTer35) variant, and case 11, who carried the c.507C > G (p.Cys169Trp) change. Both of these probands had a complex ocular phenotype, including enlarged corneal diameter, and required surgery at 1 and 3 years of age, respectively. Finally, one proband (case 7, who had high myopia, spherophakia and soft dysmorphic features) was found to have compound heterozygous changes in *ASPH*. This gene is associated with Traboulsi syndrome, a condition characterised by facial dysmorphism, lens dislocation, anterior segment abnormalities and, in a minority of cases, cardiovascular abnormalities [6]. Cardiovascular assessment in this proband was unremarkable.

## 4. Discussion

Ectopia lentis has variable severity and requires multidisciplinary management. Here, we studied the more severe end of the ectopia lentis spectrum, as we focused on paediatric cases that required lens surgery. Our observations provide further evidence supporting a central role of genetic testing in the care pathway of these children.

Conservative management is advocated for patients who have uncomplicated ectopia lentis and good vision in order to avoid loss of accommodation and surgical/anaesthetic risks [2]. Although most centres restrict surgical intervention to individuals with significant visual loss (usually 0.5 LogMAR or less) [7,8,9], a range of practices and pre-operative visual acuity thresholds have been described in the biomedical literature [7,10]. It is highlighted that other disturbances of visual function, such as monocular diplopia and glare or impending total lens displacement may necessitate surgical correction [7,8]. In this cohort, the pre-operative corrected visual acuity ranged between 0.20 and 1.28 LogMAR; there was improvement post-operatively in 17/22 eyes to an average of 0.32 LogMAR. In general, visual outcomes and complication rates were in keeping with other series (Table 4) [8,9,10,11,12,13], although direct comparisons are not possible due to the observed heterogeneity in pre-operative vision and age at intervention.

The most common cause of non-traumatic ectopia lentis in many populations is Marfan syndrome [14,15]. Children with this condition require regular cardiovascular screening. It is, therefore, important to consider the possibility of a Marfan syndrome diagnosis in all paediatric patients with lens dislocation [16]. However, other important causes of ectopia lentis have been described [1,3]. Conditions in which disease manifestations are confined to the eye (including isolated ectopia lentis and disorders where ectopia lentis is combined with other eye abnormalities) affect a significant proportion of cases. Such “ocular” forms of lens dislocation accounted for 46% (5/11) of participants in this cohort; this is comparable with the findings of other, similar studies [1,14]. Notably, inborn errors of metabolism are recognised as a rare cause of ectopia lentis. In a large study from Denmark, such conditions accounted only for 4% (7/396) of individuals with ectopia lentis [14]. It is therefore, unsurprising that none of the probands in this small-scale study had a metabolic disorder. It is noted that inborn errors of metabolism are typically diagnosed early in life. For example, sulphite oxidase deficiency is often identified during the neonatal or infantile period [17], while homocystinuria is often picked up through newborn screening programs [18]. The median age at presentation in this cohort was 3 years, and the youngest proband presented in infancy.

**Table 4 genes-14-00791-t004:** Comparison of surgical procedures and outcomes between select case series of surgical management of paediatric ectopia lentis.

Study (Cases Undergoing Surgery)	Age at Time of Surgery (Years)	Surgical Technique	IOL Implantation	Post-Operative Surgical Complications	Pre-op BCVA LogMAR	Final Post-op BCVA LogMAR
**Current study (*n* = 22)**	5.0 (median)	Anterior approach (*n* = 18); Posterior approach (*n* = 4) §	Aphakia	Vitreous haemorrhage (*n* = 1); Glaucoma (*n* = 4)	0.54 (median)	0.22 (median)
**Anteby et al.** [8] **(*n* = 38)**	6.4 (mean)	Anterior approach (*n* = 11); Posterior approach (*n* = 27)	Aphakia	Vitreous haemorrhage (*n* = 1); Retinal detachment (*n* = 1); Retinal tear *n* = 1	0.92 (mean)	0.20 (mean)
**Wu-Chen et al.** [9] **(*n* = 17)**	7.7 (mean)	Anterior approach (*n* = 1); Posterior approach (*n* = 16)	Aphakia (*n* = 17; scleral fixated lens 6 years later in one case)	Glaucoma (*n* = 1); Transient ocular hypertension (*n* = 1); Vitreous haemorrhage (*n* = 1); Partial PVD *n* = 2	0.7 (median)	0.1 (median)
**Konradsen et al.** [10] **(*n* = 37)**	4.3 (median)	Anterior approach (no anterior vitrectomy, posterior capsule intact)	Acrysof IOL (Alcon) in the capsular bag	Visual axis opacification (*n* = 31); IOL dislocation (*n* = 2); Suture-related discomfort (*n* = 2); Anterior synaechia (*n* = 1)	0.59 (median)	0.23 (median)
**Català-Mora et al.** [11] **(*n* = 21)**	8.0 (mean)	Posterior approach	Artisan (iris-claw) IOL	Vitreous haemorrhage (*n* = 6); Anterior uveitis (*n* = 1); IOL dislocation (*n* = 1); Retinal detachment (*n* = 1); Macular oedema (*n* = 1)	0.91 (mean)	0.18 (mean)
**Kopel et al.** [12] **(*n* = 22)**	6.5 (mean)	Posterior approach	Aphakia (*n* = 10); Iris-sutured 3-piece IOL (*n* = 12)	IOL dislocation (*n* = 4)	0.81 (aphakic group); 0.83 (IOL group) (mean)	0.41 (aphakic group) 0.24 (IOL group) (mean)
**Cai et al.** [13] **(*n* = 101)**	6.1 (mean)	Anterior approach (posterior capsulectomy and anterior vitrectomy if <5 years)	IOL in the capsular bag	PCO (*n* = 55); Elevated IOP (*n* = 3); Vitreous haemorrhage (*n* = 2); Posterior synechiae (*n* = 3); IOL dislocation (*n* = 4)	0.68 (mean)	0.10 (mean)
**Van Hoorde** [19] **(*n* = 2)**	17.5 (mean)	Anterior approach (posterior capsulorrhexis in one, anterior vitrectomy in the other)	Aphakia	Nil reported	0.6 (mean)	0.95 (mean)

§ Anterior approach involves a corneoscleral incision, anterior capsulorhexis, lensectomy, removal of the lens capsule in its entirety, anterior vitrectomy and peripheral iridectomy. Posterior approach refers to lensectomy via pars planar incision and vitrectomy; a corneal incision is often constructed to facilitate insertion of the intraocular lens. Visual acuities reported in Snellen or other formats were converted to LogMAR for comparison. BCVA, best corrected visual acuity; IOL, intraocular lens implant; PVD, posterior vitreous detachment; PCO, posterior capsular opacification.

Establishing the diagnosis of Marfan syndrome clinically can be challenging (especially in relatively mild or atypical cases), and many patients remain undiagnosed until later in life [4,20]. In the absence of a formal diagnosis, the management of individuals with features of Marfan syndrome (including cases with ectopia lentis, even when this is apparently/presumably isolated) should be approached with caution. For example, there are general anaesthetic considerations that may affect surgical planning in Marfan syndrome suspects [20,21,22]. Additionally, regular monitoring/screening is indicated, and this has been shown to contribute significantly to the burden of care experienced by affected individuals [23]. It is highlighted that Marfan syndrome-related cardiovascular defects have been detected in cases with previously presumed isolated ectopia lentis [4]. 

Five children with *FBN1*-associated ectopia lentis (presumed Marfan syndrome) were identified in this cohort. One of them had a synonymous change, another had a splice-site variant, one carried a missense change altering a cysteine residue, one had deletions of multiple exons, and one had an insertion-deletion that impacted a splice junction (Table 3). Several studies have attempted to identify genotype–phenotype correlations and to link *FBN1* variant classes with risk of ectopia lentis. For example, a group from Germany described findings in 587 children with Marfan syndrome. *FBN1* variants were identified in 65% of cases (using high-throughput sequencing). Differences in ocular and cardiovascular manifestations were noted among different genotype groups, and ectopia lentis was reported to be more common in Marfan syndrome patients that carry missense variants (and especially in those with cysteine-involving missense changes) [24]. In another study, a group from China described a cohort of 123 individuals with Marfan or Marfan-like syndromes. A genetic diagnosis was identified in 79% of cases. Ectopia lentis was present in 34 probands in this cohort, and genotype-phenotype correlation analysis did not find a significant difference in ectopia lentis rates between a splice site and a missense mutation group [25]. 

It can be extrapolated from this study is that gene panel testing can be used to stratify children with ectopia lentis into high- and low-risk groups for extraocular complications (Figure 2). Genetic analysis in this small-scale cohort suggested that cardiovascular manifestations are less common in children who have non-traumatic ectopia lentis and: require lensectomy in the first 3 years of lifehave unaffected parents (this is unsurprising, as Marfan syndrome is classically inherited as an autosomal dominant trait)initially present to ophthalmology services (e.g., prior to clinical genetics or cardiology clinics).

In this observational study, we sought to gain insights into the clinical utility of genetic testing in paediatric ectopia lentis. The ideal study design for assessing the value of an investigation/intervention is a randomized controlled trial. Although this retrospective case series was experimentally less rigorous than a prospective randomized trial, it allowed collection of notable findings, and it is representative of real-world clinical care [5]. Notably, there is no consensus on what constitutes sufficient evidence to justify implementation of genetic testing. It can however be argued that, on certain occasions, alignment between observational studies and mechanistic reasoning can reduce the need for prospective trials.

Overall, we have shown that panel-based genetic testing can help obtain a precise diagnosis and can guide ongoing management in children at the severe end of the ectopia lentis spectrum. We propose the early introduction of genetic analysis in affected probands, including in children who have no family history. 

## Figures and Tables

**Figure 1 genes-14-00791-f001:**
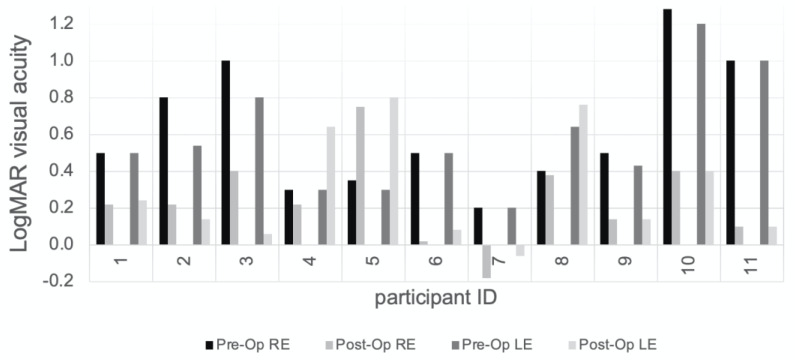
Pre-operative and post-operative LogMAR (or measurement converted into LogMAR equivalent) corrected visual acuity for a cohort of children with non-traumatic ectopia lentis. The pre-operative visual acuity achieved with both eyes open is shown for cases 1, 4 and 6. Further information including numerical data can be found in Table 2. RE, right eye; LE, left eye.

**Figure 2 genes-14-00791-f002:**
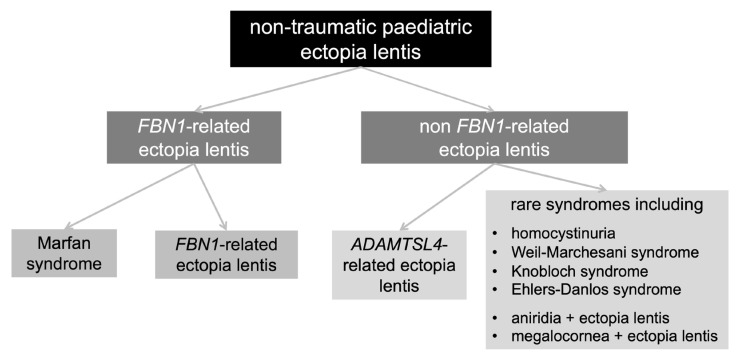
Main subtypes of non-traumatic paediatric ectopia lentis. It is noted that all children with *FBN1*-related ectopia lentis require surveillance for extraocular complications (including aortic root disease).

**Table 1 genes-14-00791-t001:** Clinical characteristics in a cohort of paediatric ectopia lentis.

Case ID	Sex	Broad Ancestral Group	Presenting Symptom	Initial Presentation to Ophthalmology?	History of Consanguinity	Family History of Ectopia Lentis or Marfan Syndrome	Age at Presentation (Years)	Age at Time of First Lens Procedure (Years)	Age at Genetic Testing (Years)
**1**	Male	White European	Iris flickering	Yes	No	No	<1	2	1
**2**	Female	South Asian	Reduced vision	Yes	Yes	Yes	3	5	5
**3**	Male	White European	Iris flickering	Yes	No	No	1	3	5
**4**	Female	South Asian	Iris dislocation noted following minor trauma	Yes	Yes	No	1	1	1
**5**	Female	White European	Iris flickering	Yes	No	No	4	5	4
**6**	Male	White European	Reduced vision	Yes	No	No	3	3	3
**7**	Female	White European	High myopia	Yes	No	Yes	5	8	8
**8**	Male	South Asian	Screening due to family history of Marfan syndrome	No	No	Yes	2	8	8
**9**	Female	South Asian	High arch palate at postnatal screening	No	No	Yes	3	5	3
**10**	Female	White European	Screening due to family history of Marfan syndrome	No	No	Yes	3	7	4
**11**	Female	South Asian	Reduced vision	Yes	No	No	3	3	3

**Table 2 genes-14-00791-t002:** Surgical outcomes in a cohort of paediatric ectopia lentis.

Case ID	Pre-op BCVA LogMAR	Pre-op Refraction (SE)	Axial Length (mm)	Post-op Refraction (SE)	BCVA at Last Visit (LogMAR)	Surgical Approach	Grade of Operating Surgeon	Lens Implant (Correction)	Glaucoma Present (Treatment)
**1**	0.80 R; 0.80 L; 0.50 BEO	+12.00 RE; +12.00 LE	24.01 RE; 22.86 LE	+10.50 RE; +10.00 LE	0.22 RE; 0.24 LE	Anterior approach	Consultant	Aphakia (contact lenses)	No
**2**	0.80 R; 0.54 L	−9.00 RE; −4.00 LE	24.01 RE; 22.86 LE	+9.50 RE; +12.00 LE	0.22 RE; 0.14 LE	Anterior approach	Consultant	Aphakia (glasses)	No
**3**	1.00 R; 0.80 L	−9.75 RE; −7.50 LE	20.95 RE; 21.13 LE	+14.50 RE; +15.00 LE	0.40 RE; 0.06 LE	Anterior approach	Consultant	Aphakia	No
**4**	0.30 BEO	unknown RE; +19.00 LE	19.19 RE; 19.25 LE	+21.75 RE; +19.50 LE	0.22 RE; 0.64 LE	Anterior approach	Consultant	Aphakia (glasses)	No
**5**	0.35 R; 0.30 L	−2.75 RE; −3.25 LE	22.27 RE; 22.39 LE	+13.375 RE; +13.00 LE	0.75 RE; 0.80 LE	Anterior approach	Consultant	Aphakia (glasses)	Yes (bilateral filtration surgery)
**6**	0.50 BEO	−4.25 RE; −4.25 LE	unknown	+13.25 RE; +15.00 LE	0.00 RE; 0.08 LE	Anterior approach	Fellow	Aphakia (glasses)	No
**7**	0.20 R; 0.20 L	−20.00 RE; −18.25 LE	21.22 RE; 21.91 LE	+11.75 RE; +12.875 LE	−0.18 RE; −0.06 LE	Anterior approach	Consultant	Aphakia (glasses)	No
**8**	0.40 R; 0.64 L	+5.375 RE; +6.50 LE	unknown	+4.875 RE; +5.75 LE	0.38 RE; 0.76 LE	Posterior approach	Fellow	Aphakia (glasses)	No
**9**	0.50 R; 0.43 L	−9.00 RE; −13.00 LE	22.76 RE; 22.77 LE	+12.75 RE; +13.00 LE	0.14 RE; 0.14 LE	Anterior approach	Consultant	Aphakia (glasses)	No
**10**	1.28 RE; 1.20 LE	−6,75 RE; −6.75 LE	23.89 RE; 24.06 LE	+12.125 RE; +13.00 LE	0.40 RE; 0.40 LE	Anterior approach	Consultant	Aphakia (glasses)	No
**11**	1.00 RE; 1.00 LE	−20.75 RE; +11.50 LE	22.40 RE; 22.21 LE	+9.25 RE; +9.75 LE	0.10 RE; 0.10 LE	Posterior approach	Consultant	Aphakia (glasses)	Yes (drops)

BCVA, best corrected visual acuity; SE, spherical equivalent; RE, right eye; LE, left eye; BEO, both eyes open.

**Table 3 genes-14-00791-t003:** Genetic testing findings and extraocular findings in a cohort of paediatric ectopia lentis.

Case ID	Gene	Variant and Zygosity	Other Ocular Abnormalities	Extraocular Features
**1**	*ADAMTSL4*	c.767_786del20 (p.Gln256ProfsTer8) homozygous	Ectopia pupillae (RE and LE)	None
**2**	No causal variant(s) detected	Not applicable	High myopia (RE)	Patent ductus arteriosus (corrected with cardiac catheterisation at age 3 years)
**3**	*ADAMTSL4*	c.767_786del20 (p.Gln256ProfsTer38) heterozygous; c.2236C > T (p.Arg746Cys) heterozygous	High myopia (RE and LE)	None
**4**	*LTBP2*	c.3427delC (p.Gln1143ArgfsTer35) homozygous	Increased corneal diameter, high intraocular pressure (RE and LE)	None
**5**	*FBN1*	c.6354C > T (p.Ile2118Ile) heterozygous	None	Pectus carinatum; flat feet, long toes and fingers, high-arched palate; normal cardiology assessment
**6**	*FBN1*	c.356G > A (p.Cys119Tyr) heterozygous	None	Flat feet, long toes and fingers; normal cardiology assessment
**7**	*ASPH*	c.1965C > A (p.Tyr565Ter) heterozygous; c.2127-2delA heterozygous	Spherophakia, high myopia, high intraocular pressure (RE and LE)	Skeletal and facial features of Marfan syndrome; normal cardiology assessment
**8**	*FBN1*	exons 46 to 48 deletion heterozygous; exons 56 to 58 duplication heterozygous	None	None
**9**	*FBN1*	c.7204 + 1G > A heterozygous	High myopia	Skeletal and facial features of Marfan syndrome; normal cardiology assessment
**10**	*FBN1*	c.5789-9_5794del15insA heterozygous	High myopia	Skeletal and facial features of Marfan syndrome; aortic aneurysm, fenestrated atrial septal defect
**11**	*LTBP2*	c.507C > G (p.Cys169Trp) homozygous	Increased corneal diameter (RE)	None

RE, right eye; LE, left eye. The GRCh37/hg19 human genome assembly and the following transcript were used: NM_019032.4 (*ADAMTSL4*), NM_000428.2 (*LTBP2*), NM_000138.4 (*FBN1*), NM_004318.4 (*ASPH*); According to the Association for Clinical Genomic Science (ACGS) best practice guidelines for variant classification (v4.01), the following genetic changes would be classified as variants of uncertain significance (as of 02/2022):. *ADAMTSL4*: c.2236C > T (p.Arg746Cys);. *LTBP2*: c.507C > G (p.Cys169Trp);. *FBN1*: exons 56 to 58 duplication. As a result, there is a degree of uncertainty about the pathogenicity of these DNA sequence alterations.

## Data Availability

All data are contained within the article.

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
