# Peer review of "The Role of Genetic Testing in Children Requiring Surgery for Ectopia Lentis"

_genes, 2023, doi:10.3390/genes14040791_

Round 1

Reviewer 1 Report

Thank you for the manuscript, the article addresses an important and relevant subject that might interest the audience of the journal. As for the revised manuscript, I suggest more additional comments.

In line 25, “In the presented cohort..” should be revised into “Our retrospective cohort study....

What type of ectopia lentis in this study? The author should clarify non-traumatic ectopia lentis in inclusion criteria or exclusion criteria.   

Please clarify the term “systematically collected” as a supplement material

 Why this study was only tenets by Declaration of Helsinki. I think it should tenets by both Declaration of Helsinki and GCP. Moreover, the author should provide an Ethical number or approval number.   

Author Response

We would like to thank the reviewer for reviewing our manuscript. A point-by-point response can be found below (our responses are highlighted in blue font for clarity). Thank you for your comments. The following changes have been done: 

COMMENT #1.1: In line 25, “In the presented cohort..” should be revised into “Our retrospective cohort study....

RESPONSE #1.1: We have now amended the Abstract as recommended.

COMMENT #1.2: What type of ectopia lentis in this study? The author should clarify non-traumatic ectopia lentis in inclusion criteria or exclusion criteria.

RESPONSE #1.2: We have now amended the inclusion criteria and the following has now been included "a diagnosis of unilateral or bilateral non-traumatic ectopia lentis"

COMMENT #1.3: Please clarify the term “systematically collected” as a supplement material

RESPONSE #1.3: The following has now been included in the Methods section: "The case notes and electronic healthcare record entries from all study participants were inspected and relevant clinical data were systemically collected. In brief, documented encounters with Ophthalmology, Clinical Genetics and other medical specialties were reviewed, and the relevant participant information were captured using a spreadsheet application". This standardised data extraction spreadsheet was used to generate Tables 1, 2 and 3.

COMMENT #1.4:  Why this study was only tenets by Declaration of Helsinki. I think it should tenets by both Declaration of Helsinki and GCP. Moreover, the author should provide an Ethical number or approval number.   

RESPONSE #1.4: We have now expanded the relevant Methods section to include a mention to the fact that the GCP guidelines were followed throughout the research process. As this study is a service improvement / audit project, no explicit consent from participants was required. 

Reviewer 2 Report

In this manuscript, the authors present genetic analyses on retrospectively recruited patients of ectopia lentis. Based on the results from 11 probands, the authors found mutations in some of the genes previously associated with ectopia lentis. Overall, the authors suggest that patients on the extreme spectrum may benefit from genetic analyses early in the disease as it may guide clinical care. 

This manuscript is well-written and useful for the field. Only comment is that in lines 82-83, the authors list that patients with FBN1 mutations are 3/11, however, the table lists that there are 5 patients with FBN1 mutations. Please correct this.

Author Response

We thank the reviewer for his commen on our manuscript.

COMMENT #2.1: Only comment is that in lines 82-83, the authors list that patients with FBN1 mutations are 3/11, however, the table lists that there are 5 patients with FBN1 mutations. Please correct this.

RESPONSE #2.1: Indeed as the reviewer mentions, there were 5 patients with FBN1 mutations. Of these 3 presented to a different medical specialty than Ophthalmology in the first instance. We have now clarified the text to "The remaining 3 children (27%), who had input from a different clinical service in the hospital setting before having an ophthalmology review, were all found to carry genetic variants in the FBN1 gene".